# Colonization of Group B *Streptococcus* in Pregnant Women and Their Neonates from a Sri Lankan Hospital

**DOI:** 10.3390/pathogens11040386

**Published:** 2022-03-23

**Authors:** Dulmini Nanayakkara Sapugahawatte, Carmen Li, Veranja Liyanapathirana, Chaminda Kandauda, Champika Gihan, Chendi Zhu, Norman Wai Sing Lo, Kam Tak Wong, Margaret Ip

**Affiliations:** 1Department of Microbiology, Faculty of Medicine, The Chinese University of Hong Kong, Hong Kong, China; dulmini87@hotmail.com (D.N.S.); 2carmen.li@cuhk.edu.hk (C.L.); samzhu@cuhk.edu.hk (C.Z.); normanlo@cuhk.edu.hk (N.W.S.L.); kamtakwong@cuhk.edu.hk (K.T.W.); 2Department of Microbiology, Faculty of Medicine, University of Peradeniya, Peradeniya 20400, Sri Lanka; veranjacl@pdn.ac.lk; 3Department of Obstetrics and Gynaecology, Faculty of Medicine, University of Peradeniya, Peradeniya 20400, Sri Lanka; chamindakandaudagynobs@gmail.com (C.K.); champikagihan@yahoo.com (C.G.)

**Keywords:** group B *Streptococcus*, *Streptococcus agalactiae*, molecular epidemiology, genome analysis, pregnant women colonization, neonatal colonization

## Abstract

We investigated the molecular epidemiology of *Streptococcus agalactiae* (Group B *Streptococcus*, GBS) from carriage in a cohort of pregnant mothers and their respective newborns in a Teaching Hospital in Sri Lanka. GBS vaginal carriage was assessed on pregnant mothers at pre-delivery (*n* = 250), post-delivery (*n* = 130), and from peri-rectal swabs of neonates (*n* = 159) in a prospective study. All colonizing, non-duplicate GBS isolates (*n* = 60) were analyzed for antimicrobial susceptibilities, capsular serotyping, and whole-genome sequencing (WGS). The percentage of GBS carriage in mothers in the pre-delivery and post-delivery cohorts were 11.2% (*n* = 28) and 19.2% (*n* = 25), respectively, and 4.4% (*n* = 7) in neonates. GBS isolates predominantly belonged to serotype VI (17/60, 28.3%). The isolates spanned across 12 sequence types (STs), with ST1 (24/60, 40%) being the most predominant ST. Concomitant resistance to erythromycin, tetracyclines, and gentamicin was observed in eight strains (13.3%). WGS revealed the presence of antimicrobial resistance genes including *ermA* (5/60), *mefA* (1/60), *msrD* (1/60), and *tetLMO* (2/60, 28/60, and 1/60, respectively) among 60 strains. The study provides insight into the diversity of vaccine targets of GBS since serotype VI is yet to be covered in the vaccine development program.

## 1. Introduction

Streptococci were classified into many groups from A–K and H–U based on the C-carbohydrate antigen of the cell wall according to Lancefield classification [1]. *Streptococcus agalactiae* comes under Lancefield group B, thus named Group B Streptococcus or GBS [1,2]. Group B *Streptococcus* (GBS) by maternal rectovaginal colonization causes a range of maternal and perinatal diseases, including maternal infection, stillbirth, preterm delivery, and early- and late-onset sepsis in newborns [3,4,5,6]. GBS colonized mothers risk predisposition of the bacteria to the neonate during labour by vertical transmission, which is one of the factors for developing early-onset neonatal sepsis [7]. Although the Centres for Disease Control and Prevention (CDC) provided guidelines for GBS detection at 35–37 weeks of gestation with culture in selective enrichment broth, this cannot always be applied, especially in low- and middle-income settings [5]. Maternal colonization with GBS is the most important risk factor for early-onset disease (EOD) (0–6 days) in neonates. EOD was first reported in 1973 on a single infant of a mother from a cohort of 46 pregnant women with vaginal GBS colonization. The risk of EOD development in neonates born to mothers with GBS colonization was 2.17% at that time [8]. A recent review analysed the importance of intrapartum antibiotic prophylaxis (IAP) policies worldwide to prevent EOD due to GBS. Ninety-five countries out of 195 United Nations (UN) member states launched IAP policies with promising results of low EOD incidence observed in resource-rich countries regardless of risk factor approach or universal screening [7]. The use of IAP has been recommended in both risk-based and microbiological screening in the United States. The policy later changed to recommended microbiological screening with a rectovaginal swab for GBS colonization at 35–37 weeks’ gestation or among women with threatened preterm delivery and unknown colonization status, and administration of high-dose intravenous benzylpenicillin or ampicillin in labour among those GBS colonized subjects [9]. It also promotes offering IAP in broader situations including during labour. Other conditions recommended for IAP were women with GBS bacteriuria, previous infants with GBS disease, women with unknown colonization status, prolonged rupture of membranes, and maternal fever [9]. However, there have been no policies or reports for risk factor analysis followed by administration of IAP on the Sri Lankan health system so far. The estimated risk of EOD worldwide in the era of varying intrapartum antibiotic prophylaxis (IAP) among pregnant women colonized with GBS was at least 1.1% without a policy of IAP [3]. The risk of EOD decreased with the increase of regions implementing IAP [5,10]. Data platform from the World Health Organization (WHO) showed 1601 neonatal deaths reported from Sri Lanka in the year 2017, among which 0.04% of deaths were due to sepsis and other infectious conditions [11]. Timely action in pathogen screens and administration of antibiotics can help further reduce death outcomes in this cohort. Thus, monitoring maternal colonization of pathogens is an important preventive measure against newborn infection. 

The overall prevalence of maternal GBS colonization worldwide was 18% [10]. Prevalence was the highest in the Caribbean (34%) and the lowest in Melanesia (2%). Colonisation prevalence in Europe, North America, and Australia were similar (23%) [10]. In comparison, the prevalence in SouthAfrica was slightly higher than the Western countries (25%), and a lower prevalence was reported in West Africa (14%), Central America (10%), and South, South-east(14%), and East Asia (9%) [10]. The low prevalence data from Western Africa, Central America, South Asia, South-east Asia and East Asia might be due to a smaller number of publications in this region [10]. Ninety-eight percent of GBS from rectovaginal swabs consisted of serotypes Ia, Ib, II, III, and V [10]. Interestingly, variations in the prevalence of serotypes III and V were noted. Compared to an overall serotype III prevalence of 25% across the globe, Central America, South-east Asia, and a few South Asian countries, including India and Bangladesh, had about 11% prevalence of serotype III among their GBS isolates [10]. On the other hand, serotypes VI, VII, VIII, and IX appeared to be much more frequently observed in South-east(5.5%; 20/365), South (1.1%; 6/533), and East Asia (0.3%; 11/2937) [10]. However, serotype prevalence of colonized GBS in pregnant women from Sri Lanka has yet to be reported. Since there was no screening for GBS nor a protocol for IAP in Sri Lanka for prevention of GBS disease at the time of sample collection, our study’s aim was to determine the molecular epidemiology of GBS strains isolated from pregnant mothers and neonates admitted to a Sri Lankan hospital using whole-genome sequencing. It also aimed to provide a descriptive analysis on the maternal and neonatal colonization of GBS to address the carriage rate and serotype prevalence in Sri Lanka in which this information has been lacking to curb hospital policies in the locality.

## 2. Results

Sixty GBS strains were identified in 28 pregnant mothers on admission to the antenatal ward, 25 mothers discharged from the postnatal ward, and seven neonates who were born to the mothers that screened for maternal GBS colonization. Colonization rates were 11.2% (28/250), 19.2% (25/130) and 4.4% (7/159) in pre-delivery mothers (pre-delivery vaginal swabs), post-delivery mothers (post-delivery vaginal swabs) and neonates (neonatal anal swabs), respectively. Among them only two mothers had pre-and post-delivery swabs with GBS positive results (Isolates E33a2 & E33b6, E10a4 & E10b7) and one mother-baby paired sample were GBS positive (Isolates E3b7 & 3c10). 

### 2.1. Serotype and Multilocus Sequence Typing (MLST) Distribution

A total of seven serotypes were identified in this study (Table 1). The predominant serotype was VI, accounting for 28.3% (17/60) of all isolates, followed by serotype III (15/60, 25%), serotype II (12/60, 20%), serotype Ia (7/60, 11.7%), serotype V (6, 10%), serotype Ib (1, 1.7%), and serotype IV (1, 1.7%). The remaining strain was non-typeable (NT). Serotypes II, III, and VI were commonly found in pre-delivery vaginal swabs (21/28, 75%), while serotypes VI and III were common in post-delivery vaginal swabs (18/25, 72%). Serotypes from the seven neonatal anal swabs were Ia, VI, II, and V. MLST analyses revealed 12 different sequence types (STs) among our GBS isolates with ST1 as the predominant ST (24/60, 40%) followed by ST23 (10/60, 16.7%) (Table 2). ST1 strains mainly belonged to serotype VI (16/24, 66.7%) and ST23 isolates were mainly identified as serotype III (7/10, 70%). Over half of the CC19 isolates (including ST19, ST28, ST49, ST335) were also serotype III (six out of 10 of strains). Four strains showed allelic profiles that have not been previously reported, thus regarded as Unknown ST and belonged to serotypes II and IV.

### 2.2. Antimicrobial Susceptibility

The antimicrobial susceptibility of the 60 GBS isolates was tested against 19 antimicrobial agents (Table 3). All isolates were susceptible to penicillin, oxacillin, cefotaxime, ceftibuten, vancomycin, linezolid, chloramphenicol, clindamycin, lincomycin, ciprofloxacin, levofloxacin, gatifloxacin, moxifloxacin and all strains were resistant to gentamicin. Resistance to doxycycline, minocycline, and tetracycline was observed in 34 (56.7%), 34 (56.7%), and 35 (58.3%) of the 60 strains, respectively. Nine strains (15%) showed resistance to erythromycin, whereas eight of them also had resistance to tetracyclines and gentamicin. Twenty-eight GBS strains that showed resistance to tetracycline group antimicrobials carried tetracycline resistant genes according to whole-genome analysis data (Figure 1). Similarly, four of the erythromycin resistant strains possessed either *ermA* or *mefA* genes in their genomes. 

### 2.3. Genome-Based Phylogenetic Analysis

Whole-genome sequencing (WGS) analysis revealed that the Sri Lanka GBS strains were segregated into three clades with one of the clades containing two main clusters (Figure 1). The first clade (Clade A) consisted of 11 GBS strains serotypes II and III of CC19 (STs 19, 28, 49, and 335). The second clade (Clade B) mainly consisted of serotypes III and Ia, ST23 and 24. The third clade (Clade C) was largely ST1 and consisted of two clusters: one cluster (Clade C.1) was all serotype VI, while the other cluster (Clade C.2) was serotype II/V. Two vaginal swabs isolate pairs (E10a4 and E10b7; E33a2 and E33b6) and one mother-neonate isolate pair (E3b7 and E3c10) showed different serotypes and ST from each other. E10 pair clustered together, E33 vaginal swab pair and E3 mother-neonate pair were in different clusters (Figure 1).

**Figure 1 pathogens-11-00386-f001:**
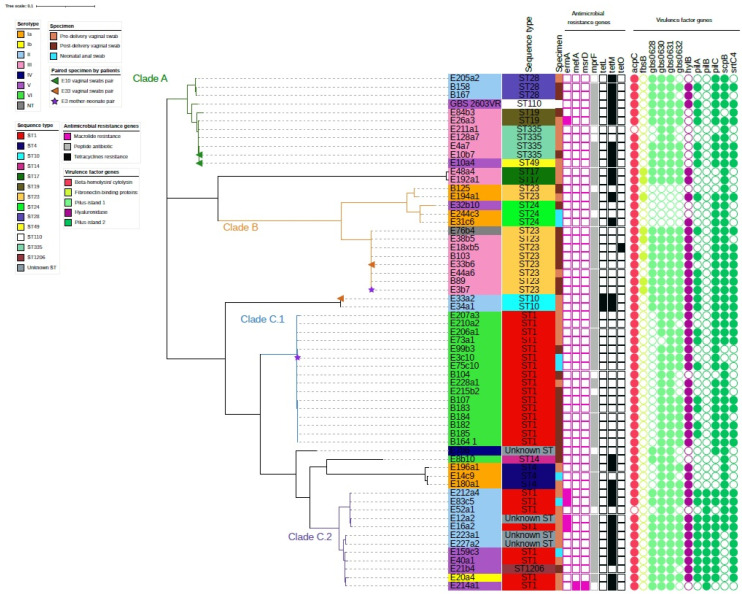
**Phylogeny of 60 *Streptococcus agalactiae* isolates and their molecular characteristics.** Serotypes of strains are noted as the color band on the sample names. Sequence types (ST) and specimen types were noted next to the sample names. Specimen type, presence of antimicrobial genes (color filled squares), presence of virulence factor genes (color filled circles) were noted in the key legend in the figure. Paired specimens of patients were noted at the end of the nodes accordingly by colored triangles for vaginal swabs pair and by colored stars for mother-neonate pair. Clades A (green branch), B (orange branch), C.1 (blue branch) and C.2 (purple branch) were noted on the tree.

### 2.4. Virulence Genes and Antibiotic Resistance Genes among the Three GBS Clusters

Antimicrobial resistance gene was scarce among our GBS cohort, while virulence genes were evident. The first clade (Clade A) contained antimicrobial gene (AMRG) *tetM* but showed the absence of virulence genes encoding fibronectin-binding proteins (*fbsB*) and hyaluronidase gene (*hylB*). AMRG *tetM* and *tetO* was only present in three out of thirteen strains in the second clade (Clade B). Serotype III ST23 is known to be a virulent strain with *hylB* present in all but one strain. *FbsB* gene was uniquely present in four out of seven strains serotype III ST23 strains. This presents a potential risk of transmission to neonates in developing GBS disease. Serotype Ia strains in this clade did not contain hypothetical genes encoding Pilus-island 1 (*gbs0628*, *gbs0630*, *gbs0631*, and *gbs0632*) and *fbsB*. Clade C.1 containing serotype VI ST1 was absent of AMRGs except for *MprF*, an AMRG which was common throughout our strains. Of note, the cluster containing serotype II/V ST1 (Clade C.2) had five strains of serotype II (E83c5, E212a4, E52a1, E16a2, and E12a2) that were previously categorized as concomitantly resistant to erythromycin, tetracyclines and gentamicin. AMRG *ermA* and *tetM* were also found among these strains. This clade also contained unique genes encoding virulence factors related to Pilus-island 2 (*pilB*). Eight strains were concomitantly resistant to erythromycin, tetracyclines and gentamicin as observed from the antibiotic susceptibility test, where five of them had been mentioned. The remaining three strains were serotype V ST1 (E214a1), and serotype III ST335 (E211a1 and E10b7).

## 3. Discussion

Maternal colonization of GBS is an important investigation due to its potential in transmission to offspring during intra- and post-partum, leading to severe infection and eventually death. It has been postulated that GBS colonization and infection in the mother could lead to preterm birth and stillbirth. Neonates who have been survived from lethal GBS infection may still suffer from neurological impairments which required long-term care. The global maternal colonization rate of GBS is 25%, yet this data is largely derived from developed countries. However, such data in Southeast Asia is lacking and has been estimated to be 11% from a meta-analysis [10]. To the best of our knowledge, our study is the first whole-genome sequence study of Sri Lankan *Streptococcus agalactiae* isolates. This is also the largest GBS study cohort performed in Sri Lanka. The detected colonization rates of GBS among pre-delivery mothers andpost-delivery mothers were 11.2% and 19.2% respectively, while neonatal GBS colonization was 4.4%. All the mothers and neonates were healthy upon discharge. A discrepancy of 18% and 49% of GBS colonization rate was noted in Sri Lanka through different GBS detection methods (culture method Vs real-time PCR from the direct sample) from 100 pregnant women, but the study did not characterize the isolates further [13]. Our data largely concurred with a 50-subject cohort investigation on GBS colonization in Sri Lanka where a 10% colonization rate was observed [14]. Our maternal colonization rate for pre-delivery swabs was also similar to the global data [7,10]. Among our 60 GBS colonizing strains from mothers and neonates, there was a predominance of serotype VI followed by serotype III, II, Ia and V. Detection of serotype VI was also noted in neighbouring Bangladesh. This indicates the emergence of GBS serotype VI in regional countries [15]. Prevalence of ST1 encompassing serotypes VI, II and V was observed. MLST for a global GBS collection identified STs 1 and 19 to be significantly more associated with asymptomatic colonization and ST-23 was common for carriage and invasive and our results were also similar to global data [16]. Several sequence types (including ST17, ST19, ST23, and ST335) were serotype III. Serotype III is mostly associated with invasive diseases and is the overall prevalent serotype for maternal colonization across the globe with the geographical variation of lower prevalence in Central America (11%), Southeast Asia (12%) and South Asia (11%) [10,17,18]. Serotype III is less frequently found in Asia but less common serotypes VI, VII and VIII are more frequently found and indicates the emergence of these less common GBS serotypes in the region [10]. However, the recent publication of serotypes in Asia has reflected a change of prevalence. The colonization rate of GBS in Japan was 22.7% in mothers and 8.8% neonates among a cohort of 250, showing serotype prevalence of Ib followed by III, V and VI [19]. The colonization rate in Delhi, India was 15% from a cohort of 300 pregnant women where serotype prevalence was III, followed by V, II, Ia, VII and Ib [20]. A Bengali study also reported a 15% GBS carriage rate among 1,151 pregnant women and a 38% carriage rate in 68 babies born to GBS colonized mothers with a serotype prevalence of Ia and V in Mirzapur [15]. The maternal colonization rate was found to be 19.2% in Kathmandu with serotype III prevalence [21]. GBS maternal carriage rates of these neighbouring countries are higher than our Sri Lanka cohort and also differed from our prevalence (serotype VI), which indicates possible geographic distinctiveness. 

Antibiotic resistance is scarce among our isolates, but a small number of strains that were concomitantly resistant to gentamicin, erythromycin and tetracycline were confined to ST1 serotype II/V (8/60, 13.3%). Although this seems less important since penicillin is the first-line antibiotic for treating GBS infection and all our strains were susceptible to penicillin, erythromycin is one of the alternatives for the patient with allergy to penicillin. It was estimated that 10% of patients present such conditions, clindamycin, erythromycin, levofloxacin and vancomycin are important alternatives [22]. Macrolide resistance is on the rise since 2010 in the USA while a large survey in France showed a fall of resistance from 47 to 30% for the period 2007–2017 [23]. As erythromycin is the alternative to treating GBS infected patients who have a penicillin allergy, the resistance of the drug requires surveillance. Although 15% of erythromycin resistance was observed in our strains, they were all susceptible to clindamycin, levofloxacin and vancomycin. Erythromycin resistance rate on our GBS strains was similar to an Indian study whereas, erythromycin (1.85%) and clindamycin (5.55%) resistance were observed in 54 GBS isolates that were sampled from 250 pregnant women in Pakistan [24,25]. Over 56.7% of strains were resistant to tetracyclines while the *tetM* gene was observed in 46.7% (28/60) of overall strains. Tetracycline genes are believed to be the main driver of clonal expansion in GBS where the timeline of resistance matched to the common use of tetracycline in hospitals in 1948 [26]. It is interesting to note that relatively uniform resistance towards different generations of tetracyclines (tetracycline, minocycline and doxycycline) hasbeen observed in the current study and the predominant gene encoding the tetracycline resistance was *tetM* gene and particularly associated to strains in Clade A and C.2 (Figure 1). This encodes high level resistance to newer generations of tetracyclines including doxycycline and minocycline in *Enterococcus* spp. In GBS, *tetM* encodes tetracycline-minocycline resistance, and its widespread dissemination was reportedly due to the acquisition of Tn916-related transposons [27]. Universal screening and intrapartum antibiotic prophylaxis (IAP) of GBS carriage in pregnant women has not been installed in Sri Lanka, yet multi-drug resistance strains are observed. Thus, constant surveillance of GBS strains is required to observe the change of antibiotic resistance among Sri Lanka hospitals. 

As IAP can eventually induce antibiotic resistance in GBS, vaccine development is the optimal alternative for reducing maternal colonization and preventing transmission to neonates. The pentavalent vaccine is currently in the clinical trial and covers five serotypes (Ia, Ib, II, III, and V), while a hexavalent vaccine is in development which includes serotype IV. Although the pentavalent vaccine can cover up to 68.3% of our cohort (41/60 strains), patients carrying serotype VI GBS, which constitute 28.3% of mothers and neonates, would not be protected. Therefore, knowledge of serotype distribution in each country, especially in low- or middle-income countries especially in South Asia, will be important in tailoring effective serotype vaccines for their population as well as providing information for universal vaccine development.

However, there remain a few limitations to the study. The isolation of GBS from vaginal and peri-rectal samples was conducted according to the conventional culture-based method followed by sugar fermentation methods to confirm the species identification [23]. Thus, GBS strains that showed minimal, or no colony pigmentation have been missed while analyzing the data and that could have an impact on true GBS colonization in neonates and pregnant women in the current population. However, to overcome this issue in future studies, the adaptation of the culture based technique followed by MALDI-TOF-MS and PCR combinatorial detection and typing methods are recommended [13,28,29,30]. 

## 4. Materials and Methods

### 4.1. Bacterial Strains

GBS strains were collected as part of a prospective observational study carried out from 1 October 2015 to 3 January 2016 for investigation of pathogens that colonized the maternal vagina which could potentially lead to neonatal sepsis at that time [31]. Briefly, subjects were selected according to the convenience sampling method and each subject and corresponding neonate was assigned a subject number. Three swabs were collected at three timepoints wherever possible: (1) a low-vaginal swab from the mothers at the time of admission to antenatal ward; (2) a low-vaginal swab again from the mother but at the time of discharge from postnatal ward; (3) and lastly a peri-rectal swab from the corresponding neonate of the mother upon hospital discharge [31]. All swabs were transported in Amie’s transport medium within 2 h of collection and stored at −20 °C till further experiment [31]. Although vaginal swabs were obtained from 250 mothers at the antenatal ward, swabs were only obtained from 130 mothers at post-delivery in the postnatal ward and from 159 corresponding neonates. It has been noted that, these mothers and neonates had no sign of infection at the time of sample collection. Incomplete swabbing of the cohort was due to complications of the mother or neonate during hospitalization and missed cases as stated in the previous publication [31]. Ethical approval was obtained from the institutional ethical review committee, Faculty of Medicine, University of Peradeniya, Sri Lanka under project number 2015/EC/40. Bacterial strains were subcultured on blood agar with their identity confirmed by matrix-assisted laser desorption ionization-time of flight mass spectrometry (MALDI-TOF-MS) (Bruker Daltonics, Inc., Bremen, Germany). All confirmed GBS strains were stored at −80 °C for further analysis. 

### 4.2. GBS Serotyping and Antibiotic Susceptibility Test

GBS strains were subjected to serotyping by multiplex PCR as previously described [32,33,34]. Briefly, two to four bacterial colonies were incubated at 94 °C for five minutes in 200 µL lysis buffer (0.25% sodium dodecyl sulfate, 0.05 N NaOH) and centrifuged at 16,000 X g for five minutes before differentiating serotypes Ia, Ib, II-IX. Isolates that failed to be assigned to a serotype were noted as non-typeable (NT). GBS strains were tested against 19 antibiotics by broth microdilution according to CLSI with *Streptococcus pneumoniae* ATCC 49619 serving as the control strain [12,35]. Antibiotics included penicillin, oxacillin, cefotaxime, ceftibuten, vancomycin, gentamicin, doxycycline, minocycline, tetracycline, linezolid, chloramphenicol, erythromycin, clindamycin, lincomycin, ciprofloxacin, levofloxacin, gatifloxacin, moxifloxacin, and linezolid. Minimum inhibitory concentration (MIC) for the antibiotics was determined by visual inspection of bacterial growth in the wells.

### 4.3. Whole-Genome Sequencing of GBS Strains

Whole-genome sequencing (WGS) was performed as previously described by extracting bacterial DNA with Wizard Genomic DNA Purification Kit (Promega, Madison, WI, USA) followed by library preparation using Nextera XT Library Preparation Kit (Illumina, San Diego, CA, USA) or Riptide High Throughput Rapid Library Preparation Kit (iGenomx, Carlsbad, CA, USA) according to manufacturers’ protocol [33]. Genome sequencing was performed with NextSeq mid-output 500 System (Illumina, San Diego, CA, USA) to obtain an approximate minimum of 30 × average coverage of 150 bp pair-end sequence data. Genomes were assembled and analyzed as previously described [33]. MLST, antibiotic resistance genes (ARGs), and virulence factors were matched to PubMLST (www.pubmlst.org/sagalactiae) (accessed on 17 June 2021), MEGAres (https://megares.meglab.org/) (accessed on 17 June 2021), and VFDB, respectively, through ABRicate software (https://github.com/tseemann/abricate) (accessed on 17 June 2021). A phylogeny was constructed by PARSNP with assembled contigs while GBS 2603VR (GenBank Accession No.: NC_004116.1) (accessed on 17 June 2021) was used as a reference and the tree visualized with iTOL software (https://itol.embl.de/) (accessed on 17 June 2021). Genome assemblies of our GBS strains are available in NCBI Bio Project (No.: PRJNA739844).

## 5. Conclusions

In summary, this is the first whole-genome sequencing study to report the molecular epidemiology of GBS in Sri Lankan pregnant women and neonates. The maternal and neonatal colonization rate of GBS were 11.2% and 4.4% respectively. Prevalence of ST1 encompassing Serotypes VI, II and V were observed in the cohort. The antimicrobial resistance rate was low, but erythromycin and tetracycline resistance were localized in ST1 serotype II and V strains while ST1 serotype VI strains were all susceptible to antibiotics. Our study underlays a benchmark for the genomic characteristics of GBS in the pre-IAP era. Although we performed the largest sampling cohort, a multicenter study across regions in the country complementary with mother-neonate associated sampling would be most useful in entailing the epidemiological change of this bacteria.

## Figures and Tables

**Table 1 pathogens-11-00386-t001:** Distribution of capsular antigen types among 60 GBS isolates from vaginal swabs at pre-delivery and post-delivery of pregnant women, and peri-anal swabs from their neonates in Sri Lanka.

Serotypes	Total No. Swabs (*n* = 60)	Pre-Delivery Vaginal Swab (*n* = 28)	Post-Delivery Vaginal Swabs (*n* = 25)	Neonate Anal Swabs (*n* = 7)	*p*-Value (Pre- vs Post-Delivery Vaginal Swabs) ^^^	*p*-Value (Pre-Delivery Vaginal Swabs vs Neonatal Anal Swabs) ^^^	*p*-Value (Post-Delivery Vaginal Swabs vs Neonatal Anal Swabs) ^^^
Ia	7 (11.7%)	3 (10.7%)	1 (4%)	3 (42.9%)	0.6	0.07	**0.03**
Ib	1 (1.67%)	1 (3.57%)	0 (0%)	0 (0%)	N/A	N/A	N/A
II	12 (20%)	9 (32.1%)	2 (8%)	1 (14.3%)	**0.04**	0.6	1
III	15 (25%)	7 (25%)	8 (32%)	0 (0%)	0.76	N/A	N/A
IV	1 (1.67%)	0 (0%)	1 (4%)	0 (0%)	N/A	N/A	N/A
V	6 (10%)	3 (10.7%)	2 (8%)	1 (14.3%)	1	1	1
VI	17 (28.3%)	5 (17.9%)	10 (40%)	2 (28.6%)	0.12	0.6	0.7
NT ^#^	1 (1.67%)	0 (0%)	1 (4%)	0 (0%)	N/A	N/A	N/A

Data are presented as No. (%) of isolates unless otherwise indicated. ^#^ NT: Non-typeable serotype. ^^^
*p*-value was calculated by X^2^ test or Fisher Exact test for isolate numbers less than 5. N/A, not applicable.

**Table 2 pathogens-11-00386-t002:** Relationships between sequence types and serotypes in group B *Streptococcus* isolates.

Sequence Type (ST)	Serotypes							
	Ia (*n* = 7)	Ib (*n* = 1)	II (*n* = 12)	III (*n* = 15)	IV (*n* = 1)	V (*n* = 16)	VI (*n* = 17)	NT (*n* = 1) ^#^
ST1	-	1 (100%)	4 (33.3%)	-	-	3 (50%)	16 (94.1%)	-
ST23	2 (28.6%)	-	-	7 (46.7%)	-	-	-	1 (100%)
ST335	-	-	-	4 (26.7%)	-	-	-	-
ST28	-	-	3 (25%)	-	-	-	-	-
ST19	-	-	-	2 (13.3%)	-	-	-	-
ST4	3 (42.8%)	-	-	-	-	-	-	-
ST24	2 (28.6%)	-	-	-	-	1 (16.7%)	-	-
ST10	-	-	2 (16.7%)	-	-	-	-	-
ST17	-	-	-	2 (13.3%)	-	-	-	-
ST14	-	-	-	-	-	-	1 (5.9%)	-
ST49	-	-	-	-	-	1 (16.7%)	-	-
ST1206	-	-	-	-		1 (16.7%)	-	-
Unknown STs	-	-	3 (25%)	-	1 (100%)		-	-

^#^ NT: Non-typeable serotype according to multiplex PCR protocol. Clonal complex (CC)19 includes ST19, ST28 and ST335.

**Table 3 pathogens-11-00386-t003:** Antibiotic susceptibility among 60 GBS isolates.

Class of Antibiotic	Antibiotic	MIC (μg/mL) ^a^			No. of Resistance Strains (%)
		Range	MIC_50_	MIC_90_	
Penicillins	Penicillin	2–0.0625	0.06	0.06	0
	Oxacillin	32–0.03	0.25	0.5	0
Cephalosporins	Cefotaxime	2–0.12	0.12	0.12	0
	Ceftibuten	32–0.03	16	16	0
Glycopeptides	Vancomycin	8–0.25	≤0.12	≤0.12	0
Amino glycosides	Gentamicin	4–0.25	>4	>4	60 (100%)
Tetracyclines	Doxycycline ^^^	32–1	32	>32	34 (56.7%)
	Minocycline ^^^	32–1	32	>32	34 (56.7%)
	Tetracycline	32–1	16	32	35 (58.3%)
Oxazolidinones	Linezolid	64–0.06	1	1	0
Chloramphenicol	Chloramphenicol	64–2	≤1	2	0
Macrolides	Erythromycin	4–0.12	≤0.06	1	9 (15%)
Lincosamides	Clindamycin	4–0.12	≤0.06	≤0.06	0
	Lincomycin	32–0.03	0.12	0.25	0
Fluoroquinolones	Ciprofloxacin	64–0.06	0.25	0.5	0
	Levofloxacin	32–1	≤0.5	1	0
	Gatifloxacin	4–0.25	≤0.5	≤0.5	0
	Moxifloxacin *	2–0.12	≤0.12	≤0.12	0

^a^ MIC_50_ and MIC_90_, MICs at which 50% and 90% of isolates were inhibited, respectively. MIC breakpoints for GBS were referenced to CLSI [12]. ^^^ CLSI breakpoint was taken with reference to *Enterococcus* spp. * CLSI breakpoint was taken with reference to *Streptococcus pneumoniae.*

## Data Availability

Not applicable.

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
