# Peer review of "Colonization of Group B Streptococcus in Pregnant Women and Their Neonates from a Sri Lankan Hospital"

_pathogens, 2022, doi:10.3390/pathogens11040386_

Round 1
Reviewer 1 Report
Manuscript ID: pathogens-1653639
Title: Colonization of Group B Streptococcus in pregnant women and their neonates from a Sri Lankan hospital
Authors: Dulmini Nanayakkara Sapugahawatte, Carmen Li, Veranja , Liyanapathirana, Chaminda Kandauda, Champika Gihan, Chendi Zhu, Norman Wai, Sing Lo, Kam Tak Wong, Margaret Ip *
Drs. DN Sapugahawatte and C Li et al. collected and analyzed 60 clinical isolates recovered from pregnant women and neonates in a Teaching Hospital in Sri Lanka. They performed serotying by PCR, drug susceptibility testings and WGS analysis. Although this study contains the limitation of single hospital and risk of sample collection bias, because there is few reports concerning GBS from Sri Lanka and techniques of this study are sound, this manuscript is very valuable and important. Moreover, in this study, they included oxacillin and ceftibuten into drug susceptibility tests, suggesting that they had studied well and checked the existence of GBS with reduced beta-lactam susceptibility. And this manuscript is well-written.
Comment
Table 2: The predominance of ST1 and serotype VI is characteristics of Sri Lanka. Therefore, please mention it in the Discussion session.
Author Response
Reviewer 1:
Drs. DN Sapugahawatte and C Li et al. collected and analyzed 60 clinical isolates recovered from pregnant women and neonates in a Teaching Hospital in Sri Lanka. They performed serotyping by PCR, drug susceptibility testing and WGS analysis. Although this study contains the limitation of single hospital and risk of sample collection bias, because there are few reports concerning GBS from Sri Lanka and techniques of this study are sound, this manuscript is very valuable and important. Moreover, in this study, they included oxacillin and ceftibuten into drug susceptibility tests, suggesting that they had studied well and checked the existence of GBS with reduced beta-lactam susceptibility. And this manuscript is well-written.
Comment
- Table 2: The predominance of ST1 and serotype VI is characteristics of Sri Lanka. Therefore, please mention it in the Discussion session
This point was incorporated to line 211-216 in the discussion.

Reviewer 2 Report
The manuscript is overall well-written, uses current and solid methodologies and provides information for both basic scientists and clinicians of strong epidemiological importance. Although the results may be considered a bit local and cross-sectional, they might be useful in supporting the limited epidemiological picture in Sri Lanka. Nevertheless, some concerns need to be addressed before further consideration of the paper:
Abstract:
L16: vaginal carriage was assessed
L20: and 19.2%, respectively, and …
Introduction:
At the beginning of the MS, please include a few sentences on the classification of pathogenic streptococci, based on antigen classification (A, B, C, etc.). Please consider including some of the following references:
https://www.ncbi.nlm.nih.gov/pmc/articles/PMC7781025/
https://pubmed.ncbi.nlm.nih.gov/26209784/
L41: IAP- please explain the abbreviation
L42: UN- please explain the abbreviation on first mention
L53-54: that sentence is weird, please rephrase it
The introduction should be complemented with listing the dangers and potential consequences of material GBS colonization for the neonates, pathomechanism etc.
L63-75: is there an explanation for the discrepancies in reported prevalence and serotype distribution? if so (or if not), please explain it here, or at the beginning of the discussion
Results:
L82-84: this sentence should either be at the end of the intro, or at the beginning of the discussion
Table 2. Streptococcus in italics
L119: tested against 19 antimicrobial
L122: it is interesting to note the relatively uniform resistance towards different generations of tetracycclines? are there similar cases in the literature?
ug/mL instead of ug/ml (throughout the paper)
gene names should always be in italics (throughout the paper)
Figure 1. I have a large issue with the quality of Fig 1. In my opinion, the right panel should be much larger and should be represented separately, and the associated legend should be edited under that panel. this is the major correction that should be implemented in the paper.
what was the concordance between AST and WGS results?
Discussion:
some parts of the discussion just feels like t a listing of the results of previous studies, no relationship with the present data was assessed; please make this section more crisp.
Consider including a reference on the usefulness of MALDi-TOF MS in practice for GBS screening:
https://www.ncbi.nlm.nih.gov/pmc/articles/PMC7168635/
L231: Because instead of since
Methods:
L271-276: antibiotic names should not be capitalized
Author Response
Reviewer 2:
The manuscript is overall well-written, uses current and solid methodologies and provides information for both basic scientists and clinicians of strong epidemiological importance. Although the results may be considered a bit local and cross-sectional, they might be useful in supporting the limited epidemiological picture in Sri Lanka. Nevertheless, some concerns need to be addressed before further consideration of the paper:
Comments
Abstract:
- L16: vaginal carriage was assessed
The word “performed” was replaced with the word “assessed” in line 16.
- L20: and 19.2%, respectively, and …
The word “respectively” was added to the sentence in line 21.
Introduction:
- At the beginning of the MS, please include a few sentences on the classification of pathogenic streptococci, based on antigen classification (A, B, C, etc.). Please consider including some of the following references:
https://www.ncbi.nlm.nih.gov/pmc/articles/PMC7781025/
https://pubmed.ncbi.nlm.nih.gov/26209784/
The suggested section was added to introduction in line 32-35 with the given references in the following form:
Streptococci were classified into many groups from A-K and H-U based on the C-carbohydrate antigen of cell wall according to Lancefield classification [1]. Streptococcus agalactiae comes under Lancefield group B, thus named as Group B Streptococcus or GBS [1, 2].
- L41: IAP- please explain the abbreviation
The suggested change has been incorporated to the introduction in line 47.
- L42: UN- please explain the abbreviation on first mention
The suggested change has been incorporated to line 48.
- L53-54: that sentence is weird, please rephrase it
The sentence was rephrased accordingly in line 59-61 as follow:
However, there has been no policies or reports for risk factor analysis followed by administration of IAP on the Sri Lankan health system so far.
- The introduction should be complemented with listing the dangers and potential consequences of material GBS colonization for the neonates, pathomechanism etc.
The listing dangers and potential consequences of maternal colonization for neonates and pathomechanism has been incorporated in line 37-39 in the introduction, mainly highlighting that maternal GBS colonization risk neonatal sepsis due to vertical transfer during labour.
- L63-75: is there an explanation for the discrepancies in reported prevalence and serotype distribution? if so (or if not), please explain it here, or at the beginning of the discussion
The discrepancy might be due a smaller number of publications in this region [10]. This has been incorporated into line number 78-79.
Results:
- L82-84: this sentence should either be at the end of the intro, or at the beginning of the discussion
The mentioned sentence was incorporated at the end of the introduction in line 91-93.
- Table 2. Streptococcus in italics
The legend of Table 2 has been corrected accordingly.
- L119: tested against 19 antimicrobial
The word “against” incorporated to the sentence in line 131.
- L122: it is interesting to note the relatively uniform resistance towards different generations of tetracycclines? are there similar cases in the literature?
The raised point has been discussed in line 253-260.
- ug/mL instead of ug/ml (throughout the paper)
The unit has been corrected accordingly.
- gene names should always be in italics (throughout the paper)
Format of the gene names have been corrected accordingly throughout the paper.
- a 1. I have a large issue with the quality of Fig 1. In my opinion, the right panel should be much larger and should be represented separately, and the associated legend should be edited under that panel. this is the major correction that should be implemented in the paper.
Figure 1 has been reformatted according to reviewers’ comment at their best effort. However, owing to the restriction of the software, the associated legend on the left could not be relocated. The figure legend has been revised for easier referral to legend keys within the figure.
- what was the concordance between AST and WGS results?
There was concordance between AST and WGS data and it is currently incorporated in line number 138-141 in section 2.2 as follow:
Twenty-eight GBS strains that showed resistance to tetracycline group antimicrobials carried tetracycline resistant genes according to whole-genome analysis data. Similarly, four of the erythromycin resistant strains possessed either ermA or mefA genes in their genomes.
Discussion:
- some parts of the discussion just feels like t a listing of the results of previous studies, no relationship with the present data was assessed; please make this section more crisp.
Discussion was modified for better clarity.
- Consider including a reference on the usefulness of MALDi-TOF MS in practice for GBS screening:
https://www.ncbi.nlm.nih.gov/pmc/articles/PMC7168635/
The suggestion was incorporated to discussion in line 274-281.
- L231: Because instead of since
The word “As” was included for better clarity instead of word “Since” or word “Because” in line 264.
Methods:
- L271-276: antibiotic names should not be capitalized
The requested correction made in line 312-316.
